# The Usability of Neurological Occupational Therapy Case Studies Generated by ChatGPT

**DOI:** 10.3390/healthcare13111341

**Published:** 2025-06-04

**Authors:** Si-An Lee, Jin-Hyuck Park

**Affiliations:** 1Department of ICT Convergence, The Graduate School, Soonchunhyang University, Asan 31538, Republic of Korea; iop5213@naver.com; 2Department of Occupational Therapy, College of Medical Science, Soonchunhyang University, Asan 31538, Republic of Korea

**Keywords:** clinical simulation, educational technology, large language model, virtual patient

## Abstract

**Background/Objectives**: Case-based learning is essential in occupational therapy education, but developing standardized cases requires significant resources. This study explores the usability of AI-generated case studies using ChatGPT. **Methods**: A four-stage process was applied: (1) prompt development based on existing frameworks, (2) case generation ensuring diversity and relevance, (3) expert evaluation using a 5-point Likert scale, and (4) data analysis. Five neurological cases were generated and reviewed by ten experts. **Results**: Experts rated the cases highly in clinical realism (4.22/5), information comprehensiveness (4.56/5), and educational value (4.44/5). The AI-generated cases successfully provided structured occupational therapy scenarios, assessment results, and clinical questions. **Conclusions**: Five AI-generated occupational therapy cases were developed and reviewed by occupational therapy experts to evaluate their clinical realism, comprehensiveness, and educational value. While expert feedback was favorable, the effectiveness of these cases for student learning has not yet been empirically tested.

## 1. Introduction

Occupational therapists require rapid and effective decision-making skills to facilitate clients’ occupational recovery. This process begins with clinical reasoning and assessment of clients’ functional limitations, which are crucial for treatment planning [1]. Thus, occupational therapy students undergo practice-based education and licensing examinations to verify their practical competencies [2].

Through clinical training, students apply, adapt, and integrate their knowledge, thereby developing professional competencies. This hands-on experience is essential in understanding patients and delivering occupational therapy interventions [3]. Traditionally, clinical training involves face-to-face interaction with real patients [4]. However, concerns over patient rights and safety, coupled with resource constraints, have limited students’ exposure to diverse clinical experiences [5]. To address this, case-based learning is widely used in occupational therapy education to simulate clinical scenarios, enhance comprehension, and develop clinical skills [6].

Case-based learning serves as an effective alternative to traditional clinical training, preparing students for real-world practice. However, traditional case-based learning relies heavily on manually developed cases by educators, which demands substantial time and expertise, often limiting the diversity of clinical scenarios to which students are exposed [7]. Recently, artificial intelligence (AI) has emerged as a promising solution for generating virtual cases to support clinical skill development [8]. Studies have shown the potential of generative AI models, such as ChatGPT (Chatbot Generative Pre-trained Transformer), in supporting education across various fields, including healthcare. Recent studies explored the integration of ChatGPT into medical and allied health education, highlighting its application for clinical reasoning training, virtual simulations, and personalized learning support [9,10,11,12]. However, its application in occupational therapy education remains relatively limited.

ChatGPT can generate dialogue-based case scenarios, making it suitable for virtual case development. Notably, ChatGPT has demonstrated a sufficient level of competence in occupational therapy, achieving a 60% accuracy rate on the Korean Occupational Therapy Licensing Examination [13,14]. Previous studies successfully developed case-based nursing education programs using ChatGPT, validated through expert evaluation [15]. In addition, AI-generated case studies have been increasingly adopted in healthcare education to simulate real-world patient interactions and enhance learner engagement in decision-making tasks [16]. These applications support the pedagogical design of scenario-based prompts.

Despite these advantages, AI-generated content requires rigorous validation due to inherent risks, such as hallucination phenomena [17]. The application of generative AI in case-based occupational therapy education remains limited. This study aims to develop virtual cases using ChatGPT and analyze their usability through expert evaluation, exploring the potential of AI-generated cases in clinical training.

## 2. Materials and Methods

### 2.1. Study Procedure

The study followed a four-stage approach to develop virtual cases using ChatGPT:(1)Prompt development: Reviewing previous studies to determine essential case elements, such as diagnosis, patient background, and occupational performance issues.(2)Prompt input and validation: Verifying that generated responses aligned with the intended case structures. Iterative refinement was performed to improve consistency and ensure that each case followed plausible clinical reasoning.(3)Case generation: Ensuring diversity and non-redundancy in the generated cases.(4)Expert evaluation: Assessing the usability of generated cases through expert review using a structured 5-point Likert scale focused on clinical realism, information comprehensiveness, and educational value, providing preliminary expert-based validation of the educational applicability of the cases.

This study was IRB exempt as no patient-level data were used.

### 2.2. AI-Generated Clinical Case Development

#### 2.2.1. Prompt Development

This study employed ChatGPT version 3.5 (OpenAI, San Francisco, CA, USA) via a Chrome browser. The model was configured at a temperature setting of 0.7 to allow moderate creativity while maintaining clinical relevance. The maximum token limit for each prompt was set to 2048 tokens. These parameters were selected to balance output coherence, detail, and reproducibility. Prompts were formulated based on previous studies, ensuring that scenarios included client background, occupational therapy assessment results, and clinical questions. The prompt was iteratively refined to improve case accuracy and reliability, incorporating contextual elements such as environmental, social, and motivational factors [6]. To enhance scenario details, modifications were made to explicitly guide ChatGPT to integrate various contextual aspects. The final optimized prompt demonstrated high consistency and completeness in generating virtual cases.

#### 2.2.2. Case Generation

To prevent the development of redundant cases, neurological disorder patient cases were generated using prompts, resulting in a total of five cases. To ensure consistency in structure and language across all five cases, the finalized prompt was applied without modification and used within a single ChatGPT session. This approach minimized variation in language and maintained uniformity in the presentation of clinical elements such as background scenarios, assessment results, and clinical questions. The scenarios included the client’s gender, age, diagnosis, medical history, and background information in accordance with the occupational therapy casebook commonly used in South Korea [1]. The occupational therapy assessment results comprised both subjective information from the client and objective occupational therapy evaluation data. The clinical questions were based on the scenario and assessment results, covering topics related to setting occupational therapy goals, treatment planning, and key considerations for implementing occupational therapy interventions.

### 2.3. Evaluation of ChatGPT-Generated Cases

#### 2.3.1. Evaluation Tool

The generated patient cases were evaluated based on three criteria—clinical realism, information comprehensiveness, and educational value—using a 5-point Likert scale (1: very poor, 2: poor, 3: acceptable, 4: good, 5: very good). Clinical realism refers to the extent to which the case reflects real-world clinical scenarios. Information comprehensiveness indicates whether the case provides sufficient details on the patient’s medical history, symptoms, and test results. Educational value assesses whether the case is structured to facilitate problem-solving and critical thinking. The evaluation criteria were selected based on established educational standards and expert consensus relevant to clinical case development. Although the evaluation tool was not subjected to formal psychometric validation, it was designed to provide a consistent framework for preliminary expert appraisal within an exploratory research context.

#### 2.3.2. Evaluation Process

Ten experts (5 occupational therapy professors, 5 clinical therapists) with over five years of experience evaluated five generated cases. Experts were informed in advance that the cases were generated by ChatGPT, as the aim was to assess the usability and perceived educational value of AI-generated materials. The evaluation rubric was developed by the authors based on literature and expert consensus, and it consisted of three domains. Ratings were based on a 5-point Likert scale and supported by optional qualitative feedback. Each expert reviewed case quality and provided feedback. The expert panel was composed of both academic faculty and clinical practitioners to provide a balanced evaluation of the educational relevance and clinical applicability of the AI-generated cases. The panel size is consistent with established methodological standards for early-stage usability studies and content validation research, where six to ten experts are generally regarded as sufficient to ensure the reliability and relevance of expert judgment [18].

### 2.4. Data Analysis

The general characteristics of the experts and the evaluation results in both qualitative and quantitative terms were analyzed using descriptive statistics. In addition to descriptive statistics, the inter-rater reliability of experts was assessed using the Intraclass Correlation Coefficient (ICC) to determine the consistency of rating. Data analysis was conducted using SPSS version 22.0.

## 3. Results

### 3.1. Finalized Prompt

To generate clinical cases for occupational therapy students, a series of prompts was iteratively revised using ChatGPT 3.5. The initial prompt specified that cases should include three key components: (1) scenario, detailing the client’s gender, age, and background; (2) occupational therapy assessment results, including subjective and objective evaluations; and (3) clinical questions, covering treatment goals and key considerations for therapy implementation.

The first attempt successfully generated assessment results and clinical questions but lacked sufficient background details. To enhance case depth, the Occupational Therapy Intervention Process Model (OTIPM) [6] was incorporated into the second prompt, by adding contextual elements such as temporal, environmental, social, motivational, and functional aspects. These contextual elements, drawn from the OTIPM, were intentionally incorporated to enhance the ecological validity and realism of the virtual case. However, this attempt did not significantly improve the background detail.

To further refine the scenario, a third prompt introduced explicit examples of each contextual factor (e.g., whether the client plans to transfer to another hospital or return home, environmental barriers like stairs, and the patient’s role adaptation). This attempt structured background information as a checklist rather than a cohesive narrative.

In the final version, the prompt was adjusted to integrate contextual details into a seamless narrative, rather than listing them separately. This fourth version successfully generated clinically relevant and well-structured cases, making them suitable for educational use (Appendix A).

### 3.2. AI-Generated Clinical Case

Using the finalized prompt, a total of five cases were generated: four stroke cases and one spinal cord injury case. Each case successfully included a scenario, the occupational therapy assessment results, and clinical questions. A representative case is outlined below (Table 1).

### 3.3. General Characteristics of Experts

Among the experts who participated in the usability evaluation, four were male (40%) and six were female (60%), with an average age of 38.5 years. Their institutional affiliations included universities (5 experts, 50%), rehabilitation hospitals (3 experts, 30%), and general hospitals (2 experts, 20%). The average professional experience in the field was 10.2 years. Regarding academic qualifications, four held a bachelor’s degree (40%), one held a master’s degree (10%), and five held a doctoral degree (50%).

### 3.4. Case Evaluation Results

A total of 10 experts evaluated the five cases generated by ChatGPT. The average scores for clinical realism, information comprehensiveness, and educational value were 4.22, 4.56, and 4.44, respectively, using a 5-point Likert scale. The scores ranged from 4.10 to 4.70, indicating an overall rating of “good” or better. An inter-rater reliability analysis using the ICC revealed a high level of consistency among the experts. The ICC values for each domain were 0.92 for clinical realism, 0.89 for information comprehensiveness, and 0.93 for educational value, indicating excellent agreement across all three evaluation categories. The detailed evaluation results for the five cases are presented in Table 2.

In addition to quantitative scores, qualitative feedback was collected from the expert panel to enhance understanding of case usability. The comments were thematically analyzed and categorized into three key domains: clinical realism, information comprehensiveness, and educational value.

First, regarding clinical realism, several experts noted that “the cases closely resemble the types of patients I encounter in neurorehabilitation settings. The combination of functional limitations, patient motivation, and family background felt authentic” and noted that “the narrative structure provided a realistic flow of clinical reasoning, which mirrors what we expect from actual intake assessments”.

Second, in terms of information comprehensiveness, experts commented that “each case includes sufficient background, medical details, and assessment results. The subjective and objective data were well-balanced and appropriate for educational use”, and further appreciated “the diversity of symptoms and complexity in each case, which would help students consider multiple clinical priorities”.

Lastly, under the domain of educational value, reviewers emphasized that “the clinical questions are aligned with learning objectives in occupational therapy education. They encourage both problem-solving and treatment planning”, and added that “the cases are suitable for classroom discussions, especially to stimulate clinical reasoning about goal setting, intervention strategies, and psychosocial considerations”.

These qualitative insights complement the numerical scores and suggest that ChatGPT-generated cases were not only usable but also pedagogically valuable for occupational therapy education.

## 4. Discussion

This study developed and validated neurological occupational therapy cases using ChatGPT, a large language model. Through a four-stage process, a final prompt was created, and five cases were generated. Expert evaluation confirmed that the AI-generated cases were suitable for educational use, demonstrating the feasibility of integrating ChatGPT into occupational therapy training.

ChatGPT, as a natural language processing AI, excels in understanding complex queries, generating human-like responses, and summarizing vast amounts of data. It has been widely adopted in education for personalized learning [9,12,19]. The model’s ability to assist in clinical decision-making and adapt to individualized learning needs suggests its potential as an educational tool [9]. Previous studies developed case-based education programs using ChatGPT, validating their usefulness and feasibility [13]. Consistent with previous findings, this study provides preliminary support for the potential usability of ChatGPT-generated cases in occupational therapy education. Additionally, prior research has shown that ChatGPT achieved high accuracy in answering occupational therapy licensing exam questions, indicating its capability to generate clinically relevant content [12,19].

Despite the rapid advancement of AI, its application in occupational therapy education remains limited. Clinical training is essential for developing practical skills, and case-based learning is commonly used in this context [6]. However, limited exposure to real patients and reliance on a small pool of expert-developed cases present significant challenges [20]. Moreover, LLMs can serve as educational scaffolds by allowing learners to interactively create clinical content that would otherwise require expert-level knowledge. This learner-centered approach encourages active participation and supports the development of clinical reasoning, especially when expert-reviewed feedback is provided on the generated cases. Moreover, the four-stage methodology employed in this study resulted in AI-generated cases that were information-rich, clinically realistic, and pedagogically meaningful. Prompt refinement allowed for the inclusion of contextual factors, enhancing case realism. Expert evaluation results confirmed that the cases aligned with educational goals in terms of realism, comprehensiveness, and educational value. However, these findings do not confirm the actual educational impact, as student outcomes were not assessed in this study.

While generative AI shows educational promise, there remain concerns about hallucination, factual inaccuracies, and ethical risks. In the light of such concerns, evaluating the credibility of AI-generated case content is particularly important in clinical education. Since occupational therapy students may lack the expertise to independently assess the realism or accuracy of generated cases, educator supervision remains essential to ensure safe and effective use. As such, this study emphasizes that AI-generated cases should be used under the guidance of qualified professionals, especially in formal educational settings. On the other hand, ethical issues include user autonomy, potential bias, data privacy, and the impact on educator–learner dynamics [21]. Recent studies underscore the need for ethical frameworks and structured governance in AI-assisted medical education. McCoy et al. (2025) emphasized that both faculty and students require training and clear guidelines to critically assess AI-generated content, preventing overreliance on potentially inaccurate outputs [22]. Similarly, Wiese et al. (2025) highlighted key ethical principles such as transparency and bias mitigation that should be integrated into curricula [23]. These insights reinforce the importance of not only validating AI-generated content for pedagogical soundness but also cultivating ethical literacy.

Another major concern is the reliability of AI-generated information. ChatGPT, trained on data only up to 2021, may lack current medical knowledge, limiting the accuracy of its responses [14]. Furthermore, it may provide misleading or incorrect information on legal and institutional aspects, such as medical regulations, which could negatively impact case development [12,19]. Despite these limitations, ChatGPT remains useful in contexts where updated regulatory knowledge is less essential. In more complex areas, fine-tuning with curated datasets can improve content reliability [24].

Although some people expected that integrating AI into education would require significant technical skills [25], this study demonstrates that case generation using ChatGPT is highly accessible. With a well-designed prompt, clinical cases can be generated by non-specialists, making this approach readily applicable in academic settings. Given South Korea’s high internet penetration and digital fluency, AI integration may be implemented relatively smoothly [26]. However, disparities in digital literacy among users must be addressed to promote equitable use. To ensure safe and responsible use of generative AI in occupational therapy education, ethical frameworks and practical guidelines are essential.

This study is significant in that it supports the use of AI-generated cases as a cost-effective educational resource. AI-generated case-based learning can supplement or partially replace in-person training, particularly in situations where direct patient interaction is restricted, such as during pandemics. However, this study did not directly measure student learning outcomes or evaluate the effectiveness of AI-generated content on clinical reasoning.

Despite its contributions, this study has several limitations, as it focused on evaluating AI-generated cases without assessing their practical use in learning, highlighting the need for future studies to examine whether students receive appropriate feedback when responding to clinical questions. Moreover, this study did not empirically assess the impact of AI-generated cases on enhancing students’ clinical reasoning or treatment planning abilities. Although expert evaluation confirmed the content quality of the cases, further research is required to validate their educational effectiveness at the learner level. In addition, while all patient cases were entirely fictional and no real personal data were used, ethical considerations regarding the potential misuse of AI-generated clinical scenarios warrant careful attention. AI-generated cases should be employed under expert supervision to minimize risks associated with misinformation or misinterpretation. To address potential ethical risks such as misinformation, hallucinations, and oversimplification of clinical scenarios, several safeguards should be implemented when applying AI-generated content in occupational therapy education. First, a mandatory expert review process should be established to verify the clinical realism, accuracy, and contextual appropriateness of the generated cases before their use in the classroom. Second, educators should employ an ethical checklist to screen for biases, harmful assumptions, or oversights in the AI-generated content. This checklist may include items related to patient safety, inclusivity, and clinical relevance. Third, faculty-led debriefing sessions should accompany the use of AI-generated cases to help students critically reflect on the material and to correct any misconceptions. These safeguards align with emerging best practices in AI-integrated education and are necessary to ensure responsible and pedagogically sound use of generative AI tools.

Another limitation is the lack of direct comparison between the AI-generated cases and human-authored cases. Such a comparison would clarify differences in completeness, realism, and student learning outcomes. Future research should include comparative evaluation and align it with the validated clinical documentation standard. It is also important to note that AI-generated cases should be regarded as complementary to, rather than replacements for, human-authored educational content. While generative AI offers scalability and efficiency, it may produce factually inaccurate information. Therefore, careful instructor supervision is crucial to ensuring meaningful and accurate use. These considerations are consistent with current recommendations for ethical and trustworthy AI in health professions education [27]

Additionally, the use of ChatGPT 4.0 was intentionally avoided in this study, despite its superior capabilities, in order to reflect the accessibility of the freely available tools in typical educational contexts. Future studies may explore how newer models improve case quality and educational performance. Furthermore, no external tools were utilized to compensate for ChatGPT’s limitations in Korean, which may have influenced the quality of the generated cases. Although ten experts participated, the sample size limits generalizability [18]. In addition, the functional assessment components in this study reflected core domains of neurological rehabilitation, but standardized tools, such as Functional Independence Measure, were not explicitly used. Future work should explore embedding validated assessments to increase fidelity and practical utility of AI-generated cases. Lastly, as only neurological cases were generated and evaluated, future research should expand to a wider range of medical conditions, compare AI-generated cases with real-world case studies, and explore both educator and student experiences in utilizing AI-generated content for occupational therapy education.

## 5. Conclusions

This study provides preliminary evidence supporting the applicability of AI-generated cases in occupational therapy education. However, since the cases were only evaluated by expert reviewers, their actual impact on student learning remains to be established. Although therapeutic goals and intervention strategies were included through structured clinical questions, explicit progress tracking elements were not integrated. Future research should include empirical testing with learners to validate their pedagogical impact and incorporate longitudinal components to allow for outcome evaluation and iterative treatment planning.

This research contributes to the development of cost-effective educational resources while offering a practical method for integrating AI into occupational therapy training. Future studies should investigate the real-world application of ChatGPT-generated cases in educational settings. Furthermore, establishing safety guidelines and ethical frameworks will be essential to ensure the responsible integration of AI-generated cases into occupational therapy education.

## Figures and Tables

**Table 1 healthcare-13-01341-t001:** A sample of AI-generated clinical cases.

Case Component	Examples
Scenario	Diagnosis: stroke due to middle cerebral artery infarction Chief complaint: Right hemiplegia
	Residing with family Retired with husband 2 years ago High school teacher before retirement Two children are independent Active in social activities before the onset of stroke Admitted to rehabilitation hospital Goal of returning to independent living
Occupational therapy assessment results	Subjective: Desires to independently perform basic activities of daily living Aspires to resume volunteer work Strong will to recover Anxiety due to slow recovery rate Objective: ROM: The ROM of the right upper and lower extremities is limited, especially in the shoulder and elbow MMT: MMT shows that the strength of the right upper and lower limbs is less than fair, with particularly weak strength in the hands and feet Sensation: Sensation in the right upper and lower extremities is significantly reduced Balance: Difficulty maintaining balance in sitting and standing positions ADL: Requires assistance with most activities of daily living (eating 3 points, dressing 2 points, bathing 1 point, moving around 2 points)
Clinical questions	How can an occupational therapist set occupational therapy goals? What could be short-term and long-term goals for increasing independence in ADL? What is an appropriate occupational therapy method to improve the ROM and strength of the right upper and lower extremities? How can an occupational therapist create a specific exercise program to strengthen the muscles of subject’s hands and feet? What assistive devices of tools can be used to assist with ADL? How can an occupational therapist adjust subject’s environment to live independently at home? What should an occupational therapist consider for psychological stability? How can an occupational therapist support resumption of community activities and volunteer work? How can an occupational therapist educate and support families to play a role in a rehabilitation process? What sensory rehabilitation techniques can be used to restore sensation?

ADL—Activities of daily living; MMT—Manual muscle test; ROM—Range of motion.

**Table 2 healthcare-13-01341-t002:** AI-generated clinical case evaluation.

Variable	Case 1	Case 2	Case 3	Case 4	Case 5	Average
Clinical realism (score)	4.20	4.20	4.10	4.20	4.40	4.22
Information comprehensiveness (score)	4.50	4.50	4.60	4.50	4.70	4.56
Educational value (score)	4.50	4.50	4.40	4.30	4.50	4.44

## Data Availability

The datasets presented in this article are not readily available because the data are part of an ongoing study.

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
