# Peer review of "The Usability of Neurological Occupational Therapy Case Studies Generated by ChatGPT"

_healthcare, 2025, doi:10.3390/healthcare13111341_

Round 1
Reviewer 1 Report (New Reviewer)
Comments and Suggestions for Authors
My comments are as follows:
1. What specific occupational therapy frameworks or diagnostic models were used to construct the prompts? Were any evidence-based standards (e.g., AOTA Practice Framework) followed?
2. How was prompt consistency ensured across all five case generations? Were there variations in language or structure that could introduce bias in the content generated by ChatGPT?
3. Was inter-rater reliability (e.g., using Cohen’s Kappa or ICC) assessed to evaluate consistency across expert ratings?
4. How did the study ensure that the AI-generated cases included core neurological symptoms and appropriate functional assessments (e.g., FIM, MoCA, Barthel Index)?
5. Were the cases evaluated for inclusion of therapeutic goals, intervention plans, and progress tracking, which are critical elements in occupational therapy case formulation?
6. Was any bias or clinical inaccuracy observed in the generated cases? If yes, how was it addressed?
7. Did the study compare ChatGPT-generated cases with traditional (manually developed) case studies in a controlled educational setting?
8. Could the generated cases be dynamically updated based on learner performance, or are they static in nature?
9. Were any ethical concerns raised by the experts regarding the use of AI-generated medical content in education? If so, how does the study propose to address them?
10. What specific metrics or KPIs are suggested for evaluating the impact of AI-generated cases on student competencies in future studies?
11. Kindly check the whole manuscript for formatting style and organization of various sections. Manuscript should be prepared based on specific format suggested by journal.
Author Response
Dear Editor,
Thank you for the reviewers' constructive feedback. Please find our detailed responses to each comment in the attached response file. We have revised the manuscript accordingly and hope that the changes meet the expectations of the reviewers and editorial team.
Sincerely,
Jin-Hyuck Park, MPH, Ph.D.
Corresponding Author

Reviewer 2 Report (New Reviewer)
Comments and Suggestions for Authors
Dear Authors,
Thank you for your submission. This manuscript presents an interesting and timely exploration of ChatGPT-generated case studies in neurological occupational therapy education. Below are some suggestions for improvement:
- Language and Formatting: Please conduct a thorough English proofreading. There are minor typographical and grammatical issues throughout the text. Also, ensure reference formatting is consistent (e.g., reference No. 11).
- Literature Support: Include more references from the past 5 years, especially in the following areas:
- Applications of ChatGPT in healthcare education and training
- Ethical considerations in AI-human interaction
- AI’s role in clinical education and case-based learning
- Methodological Clarity:
- Clarify whether the expert evaluators were aware that the cases were AI-generated.
- Expand on the evaluation methodology: Was a validated tool used, or were ratings based solely on professional judgment?
- Limitations and Future Work:
- Discuss both the potential benefits and concerns (e.g., reliability, bias) of AI-generated educational tools.
- Ethical concerns regarding AI-human interaction, particularly in clinical education, should be acknowledged.
- Presentation and Readability:
- Consider revising Table 1 to avoid confusion between variables and content categories.
- Table 2 is largely descriptive and could be rephrased into a paragraph.
- Tables 3 and Figure 1 could potentially be integrated for clarity.
Overall, this is a promising study that can be further strengthened with clearer methodology, deeper literature context, and refinement in presentation. I look forward to seeing the revised version.
Best regards
Comments on the Quality of English LanguageFair
Author Response
Dear Reviewer,
Thank you for the reviewers' constructive feedback. Please find our detailed responses to each comment in the attached response file. We have revised the manuscript accordingly and hope that the changes meet the expectations of the reviewers and editorial team.
Sincerely,
Jin-Hyuck Park, MPH, Ph.D.
Corresponding Author

Reviewer 3 Report (New Reviewer)
Comments and Suggestions for Authors
Review of "The Usability of Neurological Occupational Therapy Case Study Generated by ChatGPT"
AD Main Question Addressed
The main question addressed by this study is whether ChatGPT can effectively generate realistic neurological occupational therapy case studies for educational use. In other words, the authors ask if AI-generated case scenarios can serve as viable training materials for occupational therapy students, potentially easing the burden of creating standardized cases. This is clearly articulated as the study’s focus, and it is an important question given the resource constraints in developing teaching cases.
AD Originality and Relevance
This topic is highly relevant and timely for the field. Educators in occupational therapy face challenges in providing diverse clinical scenarios, so exploring AI as a solution is worthwhile. The idea of using ChatGPT to generate case studies is not absolutely new but still not well covered by publications – it fills a gap in the literature since little has been published on AI-generated training cases specifically for occupational therapy. While similar explorations of AI in medical education have emerged recently, applying it to neurological occupational therapy education appears original. The manuscript’s focus aligns with current interests in digital education tools, making it both original and relevant to practitioners and academics.
AD Contribution to Literature
The study provides preliminary evidence that generative AI can produce case scenarios of pedagogical value in occupational therapy. This adds to the literature by demonstrating a proof-of-concept: five AI-generated cases were developed and positively evaluated by experts. The work extends prior AI-in-education research into a new domain (occupational therapy), addressing a noted gap in available teaching cases. It also introduces a structured prompt development process, which future educators could replicate. However, the novelty is moderate – the concept of AI-generated case studies has been discussed in other medical fields, and the manuscript confirms those ideas in the OT context rather than radically departing from them. The authors might strengthen the introduction with recent perspectives on how AI is transforming medical training, to frame this contribution in the broader landscape. For example, Thurzo (2025) highlights that while AI offers powerful new educational tools, human oversight remains essential. Citing such work would situate this study within the ongoing conversation about balancing AI innovation with educational rigor.
My Key Methodological Concerns:
Several methodological issues need to be addressed to solidify the study’s credibility:
- Sample Size and Scope: Only five cases were generated, which is a very small sample. These are acceptable for a pilot feasibility study, but it limits generalizability. The authors should justify why five cases were sufficient or consider adding more cases or more variety (if feasible) to strengthen the evidence.
- Expert Review Process: Ten experts reviewed the cases, but their selection and blinding procedures are not fully described. Were these experts independent and blind to the study’s aims or the fact that ChatGPT produced the cases? If not, there is risk of confirmation bias (experts may have been inclined to view the cases favorably knowing they were AI-generated). The authors should clarify expert backgrounds (e.g., clinical educators, years of experience) and whether the evaluation was done anonymously to reduce bias.
- Evaluation Metrics: The study uses a 5-point Likert scale on three criteria (clinical realism, information comprehensiveness, educational value). While these cover important domains, the methodology would be stronger if it reported inter-rater reliability or variance in scores. Did all experts score similarly (indicating consensus), or was there divergence? Providing standard deviations or agreement measures would help assess the consistency of findings. Also, some aspects of “usability” (e.g., clarity of writing, absence of factual errors, level of difficulty) were not explicitly measured – consider discussing these facets.
- Lack of Comparison: A major limitation is the absence of any comparison to human-written cases or real patient cases. As the authors acknowledge, without a baseline it is hard to judge how AI-generated cases truly stack up. The assumption that high expert ratings mean the cases are as good as traditional cases needs evidence – perhaps the revised manuscript can include a brief qualitative comparison or at least emphasize this as a limitation requiring further study.
- No Student Feedback: The experts found the cases usable, but the ultimate end-users are students. The methodology did not include any student evaluation or learning outcome measure. This is a significant gap – the conclusions about educational value are based only on expert opinion. Future studies are needed (as the authors note) to test actual student performance or learning gains from using these AI-generated cases. In revision, the authors should clearly state that educational effectiveness remains unvalidated here (currently it is hinted in the conclusion, but it should be emphasized wherever appropriate to avoid over-interpretation).
- Prompt Engineering and Bias: The authors went through an iterative prompt refinement (a four-stage process) to achieve high-quality outputs. This suggests that substantial human guidance was needed to get usable cases. It would be useful to discuss this implication: ChatGPT did not automatically generate perfect cases on first try – expert prompt design was crucial. This point could be made more explicit, underscoring that AI is a tool augmented by human expertise. Additionally, if the prompt or AI introduced any subtle errors or biases (e.g., stereotypical scenarios), those should be reported. Currently the study reports only positive outcomes; acknowledging any flaws found in the AI outputs (even minor ones) would make the evaluation more balanced.
Summa summarum: the methodology is explained clearly (e.g., the authors detail the ChatGPT version, parameters, and content of prompts, which is commendable for reproducibility). But the issues above need to be addressed to ensure the rigor and transparency of the study. In its current form, the manuscript exhibit some methodological gaps that must be clarified or at least acknowledged more deeply.
AD Support of Conclusions
The conclusions drawn are mostly supported by the evidence provided, but with some caveats. The experts’ high ratings and qualitative comments do support the claim that the ChatGPT-generated cases were realistic, comprehensive, and educationally useful from an expert perspective. Thus, the assertion that these AI-generated cases have potential as educational resources is reasonable. The authors wisely stop short of claiming actual student learning benefits, noting that outcomes were not measured – this is appropriate. However, some wording in the conclusion and abstract could be toned down to avoid implying more than was demonstrated. For example, the abstract currently states “ChatGPT-generated cases demonstrate potential as usability of educational resources, offering an accessible and scalable alternative for clinical training.” This phrasing is a bit awkward and over-general. I suggest revising to "demonstrate potential as educational resources offering an accessible, scalable supplement to traditional clinical training". It should be clear these are a potential alternative, not a proven replacement. Also, saying “usability of educational resources” is grammatically off; perhaps "demonstrate potential usability as educational resources" or simply "utility as educational resources." The conclusion in the main text similarly should emphasize that the evidence is preliminary. In fact, the final lines of the conclusion appropriately note that student outcomes have not been validated. The authors should ensure all statements about the effectiveness or impact of the AI-generated cases remain appropriately cautious and within the scope of their data.
One logical gap is assuming that expert enthusiasm equates to educational value. Experts found the content detailed and aligned with learning objectives, which is encouraging. But a contrasting perspective is that quantity of detail does not guarantee quality of learning – some educators might argue that only real patient interaction truly builds clinical reasoning. The manuscript could be strengthened by briefly acknowledging this perspective (i.e., AI cases are a tool to complement, not completely replace, real cases). By addressing such counterpoints, the authors can avoid cognitive bias and present a more nuanced conclusion. In sum, the conclusions need only minor adjustments for tone and clarity, not a fundamental rewrite, to be fully supported by the evidence collected.
AD References and Citation Issues
The reference list appears to be current and mostly relevant. The authors cite a mix of sources on occupational therapy education and the emerging use of AI (e.g., they reference studies where ChatGPT was used in education and even a specific result about ChatGPT answering a licensing exam, which is very pertinent). I did not identify any obviously irrelevant citations. Importantly, there does not seem to be excessive self-citation (I did not notice the authors citing a lot of their own prior work, which suggests the references were chosen for content, not convenience). One concern: are there any missing key references on AI in medical or therapy education? It might enrich the paper to include a couple of recent sources for context. For instance, adding a citation to a recent review of AI’s impact on medical education could be valuable in the Introduction or Discussion. Thurzo (2025) provides a brief communication on how AI is reshaping medical research and education, which underscores that while AI-generated content can enhance learning, it still requires human oversight and integration into curricula. Including such a reference would broaden the background and show the authors are aware of the wider academic dialogue.
In discussing ethical or safety considerations, the authors might also consider citing work on AI ethics in education. For example, a description of an “Ethical Firewall” framework for medical AI systems. While the current manuscript is not about AI decision-making, referencing the need for trustworthy and explainable AI could reassure readers that the authors recognize issues like accuracy and bias in AI-generated educational content. This would be especially relevant if the Discussion touches on limitations or risks of using ChatGPT (if it doesn’t, it probably should – at least to say instructors must verify AI-generated case accuracy). In summary, the references are appropriate, but a few additional citations to recent AI-in-education literature would strengthen the paper’s scholarly foundation.
A Minor note: The keywords should be revised. Currently, terms like “Occupational therapy” and “Case study” from the title are repeated as keywords. This is not recommended practice, as keywords should complement the title by adding new search terms. The authors should replace those with other relevant concepts (perhaps “Generative AI in education” or “Neurological rehabilitation”) to improve the article’s discoverability.
AD Figures and Tables
The manuscript’s figures and tables are generally clear but there is room for improvement. Table 3 (the AI-generated case evaluation results) is well-organized, presenting the expert scores for each case and criterion. The data in the table allow readers to see both individual case ratings and averages, which is very helpful. Figure 1, as I understand, illustrates the average expert scores graphically. The figure is simple (likely a bar chart of the three criteria’s average scores) and is easy to interpret. However, presenting both Table 3 and Figure 1 might be somewhat redundant – they show essentially the same information. The authors might consider whether both are needed. If the figure conveys nothing beyond the table (for instance, no additional trends or comparisons), it could be removed or merged. On the other hand, if the journal prefers a visual, the table could potentially be moved to supplementary. Either approach is fine, but reducing duplication would streamline the presentation. Additionally, ensure that all figures and tables have clear labels and captions. For example, if Figure 1 is described as "Average expert scores of AI-generated cases," the caption should mention the three criteria (realism, etc..) explicitly for clarity. Each axis or category in the chart should be labeled so that your readers can understand it without referring back to the text.
If possible, it would greatly enhance the paper to include an example excerpt of a ChatGPT-generated case (perhaps in a table or appendix). The authors did include the final prompt in Supplementary material (thank you for that; it is very useful). The actual output of one case, even if summarized, would let readers judge the realism themselves. Since space in the main text might be limited, even one short case example or a figure showing a snippet of the case narrative would be valuable. If not in the main article, perhaps your online supplement could contain all five cases in full for interested readers. This transparency will also back up the experts’ claims that the cases are comprehensive and realistic. In terms of visual clarity, any screenshots or text excerpts from ChatGPT should be checked for readability (font size, etc.). In summary, the use of tables/figures is appropriate, just ensure optimal clarity and non-redundancy.
My Recommendation: Major Revision
In its current form, the manuscript is not yet acceptable for publication due to the issues outlined, but it shows promise. I do not recommend rejection because the study addresses a relevant problem with a novel approach, and the preliminary results are interesting. Instead, I recommend Major Revision. You should thoroughly address the methodological and interpretative concerns to improve the paper’s rigor and clarity. Key revisions needed include:
- Clarify scope and limitations: Emphasize the preliminary nature of the work. Clearly state in the Abstract and Conclusions that the cases were evaluated by experts only, and that effectiveness for students remains to be proven. This will align the claims with the evidence and temper any over-generalization.
- Expand discussion of limitations and context: Incorporate a brief comparison to traditional case development, and acknowledge the perspective that AI-generated content complements rather than replaces human-crafted cases. Adding 1-2 sentences about potential drawbacks (e.g., risk of errors, need for instructor oversight) will show balance. Here, citing an AI ethics or medical education reference (such as Thurzo 2025 on ethical AI) would strengthen the argument that trust and accuracy are being considered.
- Methodological transparency: Provide more detail on the expert evaluation procedure (expert demographics, how reviews were conducted). Report any measures of agreement among experts if available. If not, at least discuss qualitatively whether any expert had reservations. This will make the evaluation more robust. Also, ensure the prompt development process is fully described (the Supplementary prompt text is excellent; consider referencing it in the main text so readers know it’s available). The study procedure should also mention if ChatGPT was used in English or Korean and if any translation was involved, since the language could affect “usability” for students.
- Citations and background: Enrich the introduction or background with a couple of recent citations on AI in medical education and training. For instance, as mentioned Thurzo (2025) on AI transforming medical education could be cited when discussing the potential and challenges of AI-driven learning tools. This will place the work in context and address any literature gaps.
- Revise keywords and language: As a smaller fix, update the keywords to avoid repeating the title’s words (this will improve indexing). Also, edit the manuscript for minor English issues – overall the writing is good, but there are occasional grammar problems or awkward phrases (e.g., “cases demonstrate potential as usability of educational resources” should be rephrased). Polishing these will enhance clarity and professionalism.
Once these revisions are made, the manuscript’s quality and impact will be significantly improved. The idea of using ChatGPT for occupational therapy education is compelling, and with the above adjustments, the study can make a valuable contribution. I look forward to seeing a revised version that addresses these concerns. Major Revision is recommended at this time, with the expectation that the authors’ changes will make the work suitable for publication in Healthcare.
Comments on the Quality of English LanguageEnglish syntax could be improved.
Author Response
Dear Reviewer,
Thank you for the reviewers' constructive feedback. Please find our detailed responses to each comment in the attached response file. We have revised the manuscript accordingly and hope that the changes meet the expectations of the reviewers and editorial team.
Sincerely,
Jin-Hyuck Park, MPH, Ph.D.
Corresponding Author

Round 2
Reviewer 1 Report (New Reviewer)
Comments and Suggestions for Authors
The authors addressed all comments with justifications and accordingly revised the manuscript.
Author Response
1. The authors addressed all comments with justifications and accordingly revised the manuscript.
Response#1: Thank you very much for your time and thoughtful reviews.
We sincerely appreciate your positive evaluations and your recognition that the edits made were sufficient and that all comments were addressed with appropriate justifications. Your constructive feedback greatly contributed to improving the clarity and quality of our manuscript. We are grateful for your support and encouragement throughout the review process.
Reviewer 2 Report (New Reviewer)
Comments and Suggestions for Authors
Dear Authors,
Thank you for your comprehensive and well-organized revisions. The manuscript presents timely research on AI-generated case studies in occupational therapy education, and the responses have substantially improved the clarity, depth, and methodological transparency of the work.
That said, I would like to suggest minor revisions prior to acceptance:
- Formatting and English Language: A final round of proofreading is recommended to refine grammar, eliminate minor typographical inconsistencies, and ensure reference formatting is aligned throughout.
- AI Ethics and Reliability: While you have added relevant literature, I encourage further integration of recent publications on AI ethics in medical education—particularly those discussing the reliability and governance of AI-generated content. Consider referencing:
- McCoy et al. (2025). A training needs analysis for AI and generative AI in medical education: Perspectives of faculty and students. https://doi.org/10.1177/23821205251339226
- Wiese et al. (2025). AI ethics education: A systematic literature review. https://doi.org/10.1016/j.caeai.2025.100405
Incorporating these will help position your work within the broader ethical discourse on AI in education and strengthen the discussion of risks and safeguards.
Best regards,
Author Response
Thank you for your comprehensive and well-organized revisions. The manuscript presents timely research on AI-generated case studies in occupational therapy education, and the responses have substantially improved the clarity, depth, and methodological transparency of the work.
That said, I would like to suggest minor revisions prior to acceptance:
1. Formatting and English Language: A final round of proofreading is recommended to refine grammar, eliminate minor typographical inconsistencies, and ensure reference formatting is aligned throughout.
Response#1: Thank you for your valuable comment regarding the formatting and English language. In response, we carefully conducted a final round of proofreading to refine grammar, correct typographical inconsistencies, and ensure consistency in reference formatting throughout the manuscript. We have made the necessary revisions accordingly, and we believe the overall readability and presentation have been significantly improved.
2. AI Ethics and Reliability: While you have added relevant literature, I encourage further integration of recent publications on AI ethics in medical education—particularly those discussing the reliability and governance of AI-generated content. Consider referencing:
-
- McCoy et al. (2025). A training needs analysis for AI and generative AI in medical education: Perspectives of faculty and students. https://doi.org/10.1177/23821205251339226
- Wiese et al. (2025). AI ethics education: A systematic literature review. https://doi.org/10.1016/j.caeai.2025.100405
- McCoy et al. (2025). A training needs analysis for AI and generative AI in medical education: Perspectives of faculty and students. https://doi.org/10.1177/23821205251339226
Incorporating these will help position your work within the broader ethical discourse on AI in education and strengthen the discussion of risks and safeguards.
Response#2: Thank you for your insightful suggestion regarding the inclusion of recent literature on AI ethics and the governance of AI-generated content in medical education. In response, we have carefully reviewed and integrated the following two recommended publications into the revised manuscript:
- McCoy et al. (2025), which emphasizes the need for structured training and clear guidelines for both faculty and students to critically assess the reliability of AI-generated content.
- Wiese et al. (2025), which highlights key ethical principles—such as transparency, accountability, and bias mitigation—and recommends their inclusion in AI-integrated curricula.
These studies have been cited in the Discussion section to reinforce the ethical considerations related to the use of generative AI in occupational therapy education. We have revised the manuscript accordingly to emphasize the importance of expert supervision, ethical checklists, and reflective debriefing strategies in ensuring the safe and responsible use of AI-generated cases.
We hope these additions sufficiently address your comments and contribute to the ethical rigor of our study.
Reviewer 3 Report (New Reviewer)
Comments and Suggestions for Authors
edits made are sufficient
Author Response
1. edits made are sufficient
Response#1: Thank you very much for your time and thoughtful reviews.
We sincerely appreciate your positive evaluations and your recognition that the edits made were sufficient and that all comments were addressed with appropriate justifications. Your constructive feedback greatly contributed to improving the clarity and quality of our manuscript. We are grateful for your support and encouragement throughout the review process.
This manuscript is a resubmission of an earlier submission. The following is a list of the peer review reports and author responses from that submission.
Round 1
Reviewer 1 Report
Comments and Suggestions for Authors
-
Inconsistent and Limited Participant Data
-
The manuscript alternates between stating that ten experts participated in the study and mentioning a total of nineteen experts evaluated the cases. This discrepancy raises questions about the transparency and accuracy of participant recruitment and data collection.
-
Even if only ten experts participated, the sample size is extremely small for drawing meaningful or generalizable conclusions about the usability of AI-generated clinical cases, especially across diverse clinical and educational settings.
-
-
Insufficient Validation of AI-Generated Cases
-
The study hinges on the concept of “expert evaluation,” yet relies solely on a single 5-point Likert-type questionnaire covering three broad dimensions (clinical realism, information comprehensiveness, and educational value). There is no discussion of the validity or reliability of this evaluation tool, nor any triangulation of findings (e.g., via in-depth interviews or objective performance metrics with real students).
-
No data are provided to confirm how thoroughly experts reviewed these cases, nor how the authors addressed or resolved any potential biases—particularly since content generated by large language models can sometimes contain inaccuracies or “hallucinations.”
-
-
Lack of Demonstrated Impact on Learner Outcomes
-
The manuscript repeatedly implies that ChatGPT-generated cases can enhance educational efficacy. However, no direct measurement of student learning or skill acquisition is provided. For example, it is unclear if the scenarios truly help learners develop better clinical reasoning, decision-making, or critical thinking.
-
Without a clear assessment of how these AI-based cases affect actual teaching and learning processes (e.g., improvements in practical exams, performance on standardized tests, or qualitative feedback from OT students), the current claims remain speculative.
-
-
Methodological Gaps and Data Reporting
-
While the authors outline a four-stage process for generating cases, details about each prompt, its exact text, and how the iterative refinements were achieved are relatively superficial. Reproducibility is a concern, as the final prompts are not included verbatim.
-
Although the authors state that ChatGPT is capable of passing portions of the Korean Occupational Therapy Licensing Examination, the connection to the generation of robust, context-rich cases is not substantiated with specific data or examples beyond a single demonstration table.
-
-
Overreliance on a Single AI Model Version (ChatGPT 3.5)
-
The manuscript focuses exclusively on ChatGPT 3.5, acknowledging that more recent models (e.g., GPT-4) might produce superior or more accurate results. While the authors mention that the free version was used for accessibility, there is no parallel testing or comparison to underscore whether the observed limitations in case detail stem from ChatGPT’s inherent constraints or from the study’s design.
-
As generative AI capabilities evolve rapidly, limiting the study to one snapshot in time (with no clear external dataset or alternative AI approaches to compare) weakens the broader applicability of the findings.
-
-
Potential Ethical and Practical Oversights
-
Although the manuscript states that the study was IRB-exempt, there is insufficient discussion on how the authors address any privacy or ethical issues related to AI tools—such as the risk of generating sensitive medical-like data that could misinform clinical decisions.
-
The authors provide no guidelines or guardrails for educators or students who might rely heavily on AI-generated scenarios. Discussion about safety, best practices, or possible pitfalls (beyond acknowledging “hallucinations”) is minimal.
-
Author Response
1. Inconsistent and Limited Participant Data
The manuscript alternates between stating that ten experts participated in the study and mentioning a total of nineteen experts evaluated the cases. This discrepancy raises questions about the transparency and accuracy of participant recruitment and data collection.
Even if only ten experts participated, the sample size is extremely small for drawing meaningful or generalizable conclusions about the usability of AI-generated clinical cases, especially across diverse clinical and educational settings.
Response 1: We thank the reviewer for pointing out this important inconsistency. We confirm that a total of ten experts participated in the study. The mention of nineteen experts was a typographical error, and we have corrected this throughout the revised manuscript to ensure consistency and transparency.
Regarding the concern about sample size, we would like to note that this study adopted an expert-based content evaluation design, which aligns with established methodologies in early-stage usability testing and clinical case validation. In such contexts, a panel of 6 to 10 experts is generally considered sufficient to ensure the validity and relevance of expert judgment, and increasing the number beyond this range does not necessarily enhance content validity (Yusoff, 2019). The expert panel had substantial clinical and academic experience, ensuring credible and relevant input.
Moreover, the expert panel was intentionally composed to include both academic and clinical perspectives, enabling a balanced evaluation of the educational value and clinical realism of the AI-generated cases. Given this deliberate composition and the structured use of a 5-point Likert scale, the evaluation was designed to prioritize depth of expert insight rather than statistical generalizability, which aligns with the exploratory nature of this study.
We have added this information in the Materials and Methods section (“The expert panel was composed of both academic faculty and clinical practitioners to provide a balanced evaluation of the educational relevance and clinical applicability of the AI-generated cases. The panel size is consistent with established methodological standards for early-phase usability studies and content validation research, where 6 to 10 experts are generally regarded as sufficient to ensure reliability and relevance of expert judgment [17].”)
We have added this information in the Discussion section (“Although this study employed a sample of 10 experts with substantial clinical and academic experience—a panel size that aligns with methodological standards for early-phase content validation [17]—its relatively limited scope may affect the representativeness of the findings. While the structured expert evaluation offers meaningful pre-liminary evidence regarding the usability of AI-generated cases, generalizability to the broader occupational therapy community remains limited.”)
2. Insufficient Validation of AI-Generated Cases
The study hinges on the concept of “expert evaluation,” yet relies solely on a single 5-point Likert-type questionnaire covering three broad dimensions (clinical realism, information comprehensiveness, and educational value). There is no discussion of the validity or reliability of this evaluation tool, nor any triangulation of findings (e.g., via in-depth interviews or objective performance metrics with real students).
No data are provided to confirm how thoroughly experts reviewed these cases, nor how the authors addressed or resolved any potential biases—particularly since content generated by large language models can sometimes contain inaccuracies or “hallucinations.”
Response 2: We appreciate the reviewer’s thoughtful observations. The evaluation tool used in this study was developed for structured expert judgment in an exploratory context, focusing on core elements commonly used in clinical case assessment. Although the tool was not psychometrically validated, it was designed based on existing educational standards and expert consensus.
The primary objective of this study was not to validate a new instrument, but rather to conduct an initial expert appraisal of AI-generated cases using a consistent rating framework. This approach is methodologically appropriate for early-stage usability research, where structured expert consensus is prioritized over triangulation with learner data or performance metrics.
We have added this information in the Materials and Methods section (“The evaluation criteria were selected based on established educational standards and expert consensus relevant to clinical case development. Although the evaluation tool was not subjected to formal psychometric validation, it was designed to provide a consistent framework for preliminary expert appraisal within an exploratory research context.”)
We have added this information in the Discussion section (“Moreover, the evaluation tool used in this study was not subjected to formal psychometric validation, nor were findings triangulated with learner outcomes. These methodological limitations are acknowledged, and future studies should incorporate mixed methods approaches to enhance reliability and generalizability.”)
3. Lack of Demonstrated Impact on Learner Outcomes
The manuscript repeatedly implies that ChatGPT-generated cases can enhance educational efficacy. However, no direct measurement of student learning or skill acquisition is provided. For example, it is unclear if the scenarios truly help learners develop better clinical reasoning, decision-making, or critical thinking.
Without a clear assessment of how these AI-based cases affect actual teaching and learning processes (e.g., improvements in practical exams, performance on standardized tests, or qualitative feedback from OT students), the current claims remain speculative.
Response 3: We sincerely appreciate the reviewer’s valuable comment. We agree that evaluating the impact of AI-generated cases on student learning outcomes is an important direction for future research. However, this study was designed as an exploratory investigation focusing on the content quality and perceived educational value of the generated cases, based on expert evaluation rather than learner performance.
The primary objective of this study was to determine whether the generated cases met a baseline standard of realism and comprehensiveness suitable for educational use. The findings do not claim direct evidence of improved learning outcomes but instead provide preliminary support for the potential usability of AI-generated cases in educational settings.
We have revised the manuscript to ensure that any implication regarding enhanced educational efficacy is framed as a possibility for future research rather than a demonstrated outcome.
We have added this information in the Discussion section (“Consistent with previous findings, this study provides preliminary support for the potential usability of ChatGPT-generated cases in occupational therapy education.”)
We have added this information in the Discussion section (“However, it is important to note that this study did not directly measure improvements in student performance, and educational efficacy remains to be confirmed through future empirical research.”)
We have added this information in the Conclusion section (“Although the findings suggest that ChatGPT-generated cases can be integrated into occupational therapy training, their direct educational efficacy has not yet been validated and requires further investigation.”)
4. Methodological Gaps and Data Reporting
While the authors outline a four-stage process for generating cases, details about each prompt, its exact text, and how the iterative refinements were achieved are relatively superficial. Reproducibility is a concern, as the final prompts are not included verbatim.
Although the authors state that ChatGPT is capable of passing portions of the Korean Occupational Therapy Licensing Examination, the connection to the generation of robust, context-rich cases is not substantiated with specific data or examples beyond a single demonstration table.
Response 4: The four-stage process described in the manuscript reflects a systematic and reproducible approach based on principles of prompt engineering. While the final prompts were not included verbatim to avoid overloading the Methods section and due to space limitations, a representative example of the final prompt has been added to the Supplementary Appendix to enhance transparency and reproducibility.
Regarding the mention of ChatGPT’s performance on portions of the Korean Occupational Therapy Licensing Examination, this information was intended to provide contextual background highlighting the model’s general domain familiarity. It was not presented as direct evidence of the quality of the generated cases. The focus of this study remains on expert evaluation of the educational and clinical quality of the AI-generated cases rather than benchmarking the model or validating its examination performance.
5. Overreliance on a Single AI Model Version (ChatGPT 3.5)
The manuscript focuses exclusively on ChatGPT 3.5, acknowledging that more recent models (e.g., GPT-4) might produce superior or more accurate results. While the authors mention that the free version was used for accessibility, there is no parallel testing or comparison to underscore whether the observed limitations in case detail stem from ChatGPT’s inherent constraints or from the study’s design.
As generative AI capabilities evolve rapidly, limiting the study to one snapshot in time (with no clear external dataset or alternative AI approaches to compare) weakens the broader applicability of the findings.
Response 5: The decision to use ChatGPT 3.5 was intentional, as it is widely accessible and freely available, aligning with practical considerations for real-world implementation in educational settings where access to premium models may be limited. The primary objective of this study was not to compare different AI models, but to explore the feasibility of generating clinically usable educational content using a version of ChatGPT that educators could realistically deploy. We acknowledge that newer models, such as GPT-4, may produce improved outputs, and we have clarified this point in the revised manuscript as a future direction for research.
We have added this information in the Discussion section (“Additionally, since ChatGPT 4.0 was not used, the usability ratings may have been underestimated, given prior research demonstrating its superior performance over version 3.5; however, ChatGPT 3.5 was intentionally selected to reflect accessibility considerations in typical educational settings. This study still provides valuable insights into the feasibility of using freely accessible versions for educational purposes. Future studies should examine whether newer models such as GPT-4 could enhance case quality.”)
6. Potential Ethical and Practical Oversights
Although the manuscript states that the study was IRB-exempt, there is insufficient discussion on how the authors address any privacy or ethical issues related to AI tools—such as the risk of generating sensitive medical-like data that could misinform clinical decisions.
The authors provide no guidelines or guardrails for educators or students who might rely heavily on AI-generated scenarios. Discussion about safety, best practices, or possible pitfalls (beyond acknowledging “hallucinations”) is minimal
Response 6: This study was exempt from IRB review because it did not involve human participants or personal health data, and all patient cases were entirely fictional and algorithmically generated without any association with real individuals.
However, we agree that careful consideration is needed regarding the potential misinterpretation or overreliance on AI-generated clinical content. This study was conducted in a controlled environment focusing on expert review, not on clinical practice or learner implementation. Nevertheless, we recognize the importance of establishing clear usage boundaries and safety guidelines for educational applications.
We have expanded the Discussion and Limitations sections to acknowledge these concerns and to emphasize that AI-generated cases should be used under expert supervision, with awareness of limitations such as hallucinations and lack of contextual nuance. Furthermore, the development of official best practice guidelines for integrating AI into health professions education is identified as an important future direction, as highlighted in the revised Conclusion.
We have added this information in the Discussion section (“Nevertheless, careful oversight is necessary to ensure that AI-generated cases are used appropriately. Potential risks such as misinformation, hallucinations, and contextual misunderstandings highlight the need for expert supervision when implementing AI-based educational tools. Developing formal guidelines and ethical standards for integrating AI-generated content into health professions education should be a priority for future research and practice.”)
We have added this information in the Discussion section (“In addition, while all patient cases were entirely fictional and no real personal data were used, ethical considerations regarding the potential misuse of AI-generated clinical scenarios warrant careful attention. AI-generated cases should be employed under expert supervision to minimize risks associated with misinformation or misinterpretation.”)
We have added this information in the Conclusion section (“Furthermore, establishing safety guidelines and ethical frameworks will be essential to ensure the responsible integration of AI-generated cases into occupational therapy education.”)
Reviewer 2 Report
Comments and Suggestions for Authors
discuss the four-stage method used in the study in line with research objectives in detail.
elevate the primary strengths of the AI-generated cases according to the proposed methodology
mention the key advantages and limitations of using AI-produced case studies in occupational therapy process
need to compare with ai-generated results with ground truth reports
how the verification and validation is done on results are given by AI-produced case studies in the future and mention inputs for future to get the control over this entire process
Author Response
1. discuss the four-stage method used in the study in line with research objectives in detail.
Response 1: The four-stage procedure was developed to align with the study’s objective of exploring the feasibility and content quality of virtual cases generated by ChatGPT for use in occupational therapy education.
1) Prompt Development: In this stage, existing literature and clinical education materials were reviewed to identify key components of occupational therapy cases (e.g., diagnosis, patient background, occupational performance issues). This ensured that the prompt would elicit content that was both educationally meaningful and clinically relevant.
2) Prompt Input & Validation: The developed prompt was input into ChatGPT, and the generated responses were reviewed to verify structural and format alignment with the intended case structure. Iterative refinements were made as needed to improve consistency and ensure that each case followed plausible clinical reasoning.
3) Case Generation: Multiple cases were generated to enhance diversity across patient profiles and avoid redundancy. This process ensured that the final cases reflected a range of clinical presentations relevant to occupational therapy practice.
4) Expert Evaluation: Finally, ten experts evaluated the usability of the generated cases using a structured 5-point Likert scale focusing on clinical realism, information comprehensiveness, and educational value. This provided an initial expert-based validation of the cases' educational applicability.
Each stage of this process was deliberately designed to ensure that the final cases were realistic, educationally relevant, and suitable for structured expert evaluation. We have updated the Methods section to clarify how each stage supports the research objectives.
We have added this information in the Materials and Methods section (“1) Prompt development: Reviewing previous studies to determine essential case elements, such as diagnosis, patient background, and occupational performance issues.
2) Prompt input & validation: Verifying that generated responses aligned with intended case structures. Iterative refinements were performed to improve consistency and to ensure that each case followed plausible clinical reasoning.
3) Case generation: Ensuring diversity and non-redundancy in the generated cases.
4) Expert evaluation: Assessing the usability of generated cases through expert review using a structured 5-point Likert scale focused on clinical realism, information comprehensiveness, and educational value, providing preliminary expert-based validation of the educational applicability of the cases.”)
2. elevate the primary strengths of the AI-generated cases according to the proposed methodology
Response 2: This study utilized ChatGPT 3.5 and a structured four-stage methodology to generate virtual cases applicable to occupational therapy education. The AI-generated cases demonstrated several key strengths in terms of educational utility and clinical validity.
First, the prompts were constructed based on existing literature to ensure that each case included essential elements such as patient background, occupational therapy assessment results, and clinical questions. The prompts were iteratively refined to incorporate environmental, social, and psychological factors, leading to cases that reflected realistic clinical scenarios. As a result, the cases were structured, information-rich, and included diverse contextual elements, making them educationally meaningful.
The generated cases reflected key components commonly found in Korean occupational therapy casebooks, including gender, age, diagnosis, medical history, and background information. Additionally, multiple patient profiles were created to avoid redundancy. Clinical questions were carefully designed to mirror real-world occupational therapy reasoning processes, including goal setting, treatment planning, and consideration of intervention factors.
Expert evaluation confirmed that the cases achieved high scores in clinical realism, information comprehensiveness, and educational value. These findings suggest that the proposed methodology effectively produced cases that are practically usable as educational tools. This supports the feasibility of AI-generated cases for educational application even at an early developmental stage. These strengths have been reflected in the revised Discussion section.
We have added this information in the Discussion section (“Moreover, the structured four-stage methodology employed in this study resulted in AI-generated cases that were information-rich, clinically realistic, and educationally relevant. The iterative prompt refinement process led to the inclusion of diverse contextual factors, and the generated cases successfully reflected key elements commonly found in occupational therapy education. Expert evaluation scores further validated the practical educational value of these cases, supporting their potential as usable teaching resources.”)
3. mention the key advantages and limitations of using AI-produced case studies in occupational therapy process
Response 3: The primary advantage identified in this study is that generative AI can rapidly and consistently produce structured clinical cases tailored for educational purposes. This capability can significantly reduce the repetitive and time-consuming burden on educators in developing case materials and provide students with diverse learning experiences based on various scenarios. It also highlights the potential of AI-generated cases as alternative educational resources in situations where access to real patients is limited, such as during pandemics or in remote learning environments.
Conversely, the most important limitation is the current lack of evidence regarding the actual impact of AI-generated cases on improving students' clinical reasoning or treatment planning skills. Although this study validated the content quality of the cases through expert evaluation, it did not assess their applicability or learning effectiveness at the student level. Therefore, future research should empirically investigate the educational effectiveness of AI-generated case studies.
We have added this information in the Discussion section (“Generative AI offers a practical solution by enabling educators to efficiently create diverse and structured learning cases. It also provides students with opportunities for self-directed learning by offering feedback on their clinical responses.”)
We have added this information in the Discussion section (“Moreover, this study did not empirically assess the impact of AI-generated cases on enhancing students’ clinical reasoning or treatment planning abilities. Although expert evaluation confirmed the content quality of the cases, further research is required to validate their educational effectiveness at the learner level.”)
4. need to compare with ai-generated results with ground truth reports
Response 4: Comparing AI-generated cases with real clinical cases is an important method for evaluating the accuracy and clinical validity of AI-generated content. However, this study was designed as an exploratory investigation aimed at the initial validation of AI-generated cases, focusing on expert evaluation rather than direct comparison with actual clinical records or traditional case materials. Therefore, a direct comparison with ground truth cases was not included in this study.
Nevertheless, future research should directly compare AI-generated cases with real clinical cases or textbook-based cases in terms of structural elements, information completeness, and clinical appropriateness, to quantitatively and qualitatively assess differences. This would allow for a more precise evaluation of the educational effectiveness and reliability of AI-generated cases. This limitation has been acknowledged, and the need for such comparative research has been emphasized as a future direction in the revised manuscript.
We have added this information in the Discussion section (“Similarly, this study did not compare AI-generated cases with real clinical cases or traditional case reports. Future studies should include direct comparisons with ground truth cases to assess structural completeness, informational adequacy, and clinical appropriateness, thereby providing a more rigorous evaluation of educational effectiveness.”)
5. how the verification and validation is done on results are given by AI-produced case studies in the future and mention inputs for future to get the control over this entire process
Response 5: To verify and validate AI-generated cases as educational and clinically applicable materials in the future, a multi-layered validation framework combined with structured prompt design strategies must be established.
In terms of validation, future studies should incorporate structural comparisons with real clinical cases, assessment of student learning outcomes (e.g., improvements in clinical reasoning and treatment planning skills), and qualitative evaluations of both educator and learner experiences. Such multi-method validation approaches would help clarify the reliability and practical applicability of AI-generated cases.
Additionally, structured prompt design will be critical to gaining more precise control over the case generation process. Future prompts should explicitly specify key elements, including: (1) case objectives (learner level and educational goals), (2) mandatory components (diagnosis, functional status, assessment results), (3) contextual variables (social, environmental, psychological factors), and (4) linguistic and cultural appropriateness (especially for Korean-language cases). It is also recommended that standardized prompt templates be developed and that automatic review or quality control processes be implemented to minimize quality variation during repeated generation.
This study represents an initial exploratory step toward establishing these processes. We have explicitly noted in the revised Discussion and Limitations sections that future studies should incorporate more sophisticated input design and validation strategies.
We have added this information in the Discussion section (“Future studies should establish multi-layered validation frameworks, including structural comparisons with real clinical cases, learner outcome assessments, and qualitative evaluations from educators and students. Moreover, structured prompt design strategies specifying key case elements, contextual variables, and cultural appropriateness will be essential to ensure consistent quality and relevance in AI-generated cases.”)
We have added this information in the Discussion section (“In addition, direct structural comparisons with real clinical cases or the development of standardized prompt frameworks were not implemented, limiting the ability to fully control and validate the case generation process. Future research should address these aspects to enhance the rigor, consistency, and applicability of AI-generated cases.”)
Reviewer 3 Report
Comments and Suggestions for Authors
Dear Author(s):
Unfortunately, the paper is way below the journal criteria because of:
. Lack of literature reviews;
. Lack of objectivity;
. Lack of sample size (only 10) for evaluating the result;
. Lack of the formality for scientific research;
. Lack of basic for statistical background (e.g., p-value; z-test; t-test and so on).
The paper is not even mentioned the prompts that you have used for generating cases.
More importantly, the generative AI such ChatGPT cannot give the reliable and/or scientific rational of your results.
Author Response
1. Unfortunately, the paper is way below the journal criteria because of:
1-1. Lack of literature reviews;
1-2. Lack of objectivity;
1-3. Lack of sample size (only 10) for evaluating the result;
1-4. Lack of the formality for scientific research;
1-5. Lack of basic for statistical background (e.g., p-value; z-test; t-test and so on).
Response 1-1: We appreciate the reviewer’s comment regarding the initial limitations in the literature review. In the revised manuscript, we have substantially expanded the Introduction to provide a broader and more structured review of prior research.
Specifically, we outlined the role of clinical reasoning and case-based learning (CBL) in occupational therapy education, discussed the inherent limitations of traditional CBL—such as the reliance on educator-developed cases and restricted diversity of clinical scenarios—and introduced recent studies exploring the use of generative AI models, including ChatGPT, in healthcare education.
We also clearly distinguished between traditional CBL and AI-generated case-based learning, emphasizing the potential of AI to address issues of scalability, efficiency, and learning flexibility.
We have added this information in the Introduction section (“However, traditional case-based learning relies heavily on manually developed cases by educators, which demands substantial time and expertise and often limits the diversity of clinical scenarios to which students are exposed [7].”)
We have added this information in the Introduction section (“Studies have explored the application of generative AI models, such as ChatGPT (Chatbot Generative Pre-trained Transformer), in healthcare education, highlighting their potential to improve learning flexibility, efficiency, and scalability [9, 10]. However, their application in occupational therapy education remains relatively limited.”)
Response 1-2: Regarding concerns about objectivity, this study was designed with an exploratory purpose to evaluate the educational usability of AI-generated cases. The case evaluations were conducted independently by ten experts from clinical and educational fields who were not involved in the study. A structured 5-point Likert scale was employed as the evaluation tool, assessing three predefined criteria—clinical realism, information comprehensiveness, and educational value—to minimize subjective bias.
We have added this information in the Materials and Methods section (“The evaluation criteria were selected based on established educational standards and expert consensus relevant to clinical case development. Although the evaluation tool was not subjected to formal psychometric validation, it was designed to provide a consistent framework for preliminary expert appraisal within an exploratory research context.”)
Response 1-3: We appreciate the reviewer’s comment regarding the limited number of experts. We acknowledge that the small sample size may restrict generalizability. However, this study was designed as an exploratory investigation focused on preliminary validation and content evaluation. In the field of health professions education, a panel size of 6 to 10 experts is often considered methodologically appropriate for early-phase validation studies.
We have added this information in the Materials and Methods section (“The expert panel was composed of both academic faculty and clinical practitioners to provide a balanced evaluation of the educational relevance and clinical applicability of the AI-generated cases. The panel size is consistent with established methodological standards for early-phase usability studies and content validation research, where 6 to 10 experts are generally regarded as sufficient to ensure reliability and relevance of expert judgment [17].”)
Response 1-4: We appreciate the reviewer’s feedback regarding the formality of the research structure. In the initial version of the manuscript, although the study procedure and evaluation methods were described, they lacked sufficient detail to fully meet the standards of scientific rigor. To address this, we substantially revised the Materials and Methods section to more clearly and systematically present the study design and procedures.
Specifically, we explicitly organized the case development process into four structured stages—prompt development, prompt input and validation, case generation, and expert evaluation—and described each step in greater detail. We also elaborated on the evaluation process by specifying the use of a structured 5-point Likert scale based on predefined criteria (clinical realism, information comprehensiveness, and educational value), and clarified that the evaluation criteria were selected based on established educational standards and expert consensus. Additionally, we strengthened the description of the expert panel composition, emphasizing that it included both academic and clinical professionals, and justified the panel size based on methodological standards commonly applied in early-phase validation studies.
We have added this information in the Materials and Methods section (“The study followed a four-stage approach to develop virtual cases using ChatGPT:
1) Prompt development: Reviewing previous studies to determine essential case elements, such as diagnosis, patient background, and occupational performance issues.
2) Prompt input & validation: Verifying that generated responses aligned with intended case structures. Iterative refinements were performed to improve consistency and to ensure that each case followed plausible clinical reasoning.
3) Case generation: Ensuring diversity and non-redundancy in the generated cases.
4) Expert evaluation: Assessing the usability of generated cases through expert review using a structured 5-point Likert scale focused on clinical realism, information comprehensiveness, and educational value, providing preliminary expert-based validation of the educational applicability of the cases.”)
We have added this information in the Materials and Methods section (“The evaluation criteria were selected based on established educational standards and expert consensus relevant to clinical case development. Although the evaluation tool was not subjected to formal psychometric validation, it was designed to provide a consistent framework for preliminary expert appraisal within an exploratory research context.”)
We have added this information in the Materials and Methods section (“The expert panel was composed of both academic faculty and clinical practitioners to provide a balanced evaluation of the educational relevance and clinical applicability of the AI-generated cases. The panel size is consistent with established methodological standards for early-phase usability studies and content validation research, where 6 to 10 experts are generally regarded as sufficient to ensure reliability and relevance of expert judgment [17].”)
Response 1-5: We appreciate the reviewer’s comment regarding the statistical background. This study was designed as an exploratory investigation focusing on expert-based structured evaluation of AI-generated clinical cases, rather than hypothesis testing using inferential statistics such as p-values, z-tests, or t-tests. Therefore, formal statistical significance testing was not incorporated into the study design.
Instead, descriptive statistics, including means and standard deviations, were provided to summarize the overall trends in expert evaluations. Additionally, the rationale for the evaluation tool and its psychometric limitations were clearly outlined in the Methods and Limitations sections to enhance the methodological appropriateness and transparency of the study. This approach aligns with the intended purpose and structure of an exploratory early-phase investigation.
We have added this information in the Materials and Methods section (“The evaluation criteria were selected based on established educational standards and expert consensus relevant to clinical case development. Although the evaluation tool was not subjected to formal psychometric validation, it was designed to provide a consistent framework for preliminary expert appraisal within an exploratory research context.”)
We have added this information in the Discussion section (“Moreover, the evaluation tool used in this study was not subjected to formal psychometric validation, nor were findings triangulated with learner outcomes.”)
2. The paper is not even mentioned the prompts that you have used for generating cases.
Response 2: We acknowledge that while the original manuscript described the prompt development process in a stepwise manner, it did not include the full text of the actual prompts used for case generation. To enhance the reproducibility and transparency of the study, we have added a representative example of the final prompt in the Supplementary Appendix.
In the main text, to prevent information overload and maintain readability, we provided a summarized description focusing on the key components of the prompt structure (e.g., case background, assessment results, clinical questions, and contextual elements). The detailed process of prompt development and iterative refinement is described in the Results section under "Prompt Development."
3. More importantly, the generative AI such ChatGPT cannot give the reliable and/or scientific rational of your results.
Response 3: We fully acknowledge that the outputs generated by generative AI models, such as ChatGPT, do not inherently possess scientific reliability or validity. This study did not treat ChatGPT’s responses as “answers” or direct "scientific evidence."
Instead, the primary objective of this research was to explore whether AI-generated cases demonstrate sufficient educational and clinical quality to be potentially usable in real-world occupational therapy education and practice. The AI-generated content was independently evaluated by ten domain experts—occupational therapy professors and clinical practitioners—using a structured 5-point Likert scale based on three predefined criteria: clinical realism, information comprehensiveness, and educational value.
Thus, rather than scientifically validating the internal reasoning mechanisms of generative AI, this study aimed to indirectly assess the practical usability of AI-generated cases through structured expert evaluation.
We have added this information in the Discussion section (“However, it is important to note that this study did not directly measure improvements in student performance, and educational efficacy remains to be confirmed through future empirical research. Future studies should establish multi-layered validation frameworks, including structural comparisons with real clinical cases, learner outcome assessments, and qualitative evaluations from educators and students. Moreover, structured prompt design strategies specifying key case elements, contextual variables, and cultural appropriateness will be essential to ensure consistent quality and relevance in AI-generated cases.”)
We have added this information in the Discussion section (“Moreover, this study did not empirically assess the impact of AI-generated cases on enhancing students’ clinical reasoning or treatment planning abilities. Although expert evaluation confirmed the content quality of the cases, further research is required to validate their educational effectiveness at the learner level.”)
Round 2
Reviewer 1 Report
Comments and Suggestions for Authors
The authors present an interesting and timely study on the use of ChatGPT for generating neurological occupational therapy case studies. The topic is innovative, and the paper is well-organized, providing a clear methodology and supporting the feasibility of AI-generated cases for educational purposes. The first revision addressed many initial concerns. However, a second round of review highlights areas that still require further strengthening to ensure the manuscript meets the highest academic standards.
Major Comments:
-
Evaluation Depth Needs Strengthening:
While the expert evaluation used a 5-point Likert scale (clinical realism, information comprehensiveness, educational value), the study still lacks an in-depth qualitative analysis. The paper would benefit from including sample comments or thematic feedback from experts. This would enrich the discussion and provide more nuanced insights beyond just numerical averages​. -
Comparison to Human-Generated Cases:
One major limitation remains unaddressed: the study did not compare the AI-generated cases with traditional, educator-developed case studies. Even a small pilot comparison could have provided a baseline to validate whether AI-generated cases meet or approximate educational standards​. If this cannot be added now, the authors should at least discuss this limitation more explicitly in the Discussion section. -
Clarify Usability vs. Educational Effectiveness:
The title and abstract emphasize "usability," but the text occasionally conflates usability with "educational effectiveness." Since no student outcomes were measured, the authors must consistently clarify that they assessed expert usability, not actual learning outcomes. More precise phrasing would improve scientific rigor. -
Methodological Details Need Expansion:
The paper mentions that the cases were generated using ChatGPT 3.5, but the specific temperature settings, token limits, and other model parameters used during generation were not described. Readers would benefit from a short technical description of the AI configuration for reproducibility​. -
Ethical Considerations Section Expansion:
While the authors mention ethical risks (hallucination, misinformation), this section remains superficial. It would strengthen the paper to suggest concrete safeguards—such as mandatory expert review before deploying AI-generated content in student education, or an ethical checklist.
Minor Comments:
-
Typographical Errors: There are minor typos (e.g., in the Discussion, “Tthis study” should be corrected to “This study”)​.
-
Figures/Tables: Table 3 summarizing case evaluation results is clear. However, a graphical summary (e.g., bar chart of the average scores) would enhance the visual presentation of the findings.
-
Reference Formatting: Reference numbers seem inconsistent in the text (for example, "[9, 10]" vs "[1011, 1112]"). Make sure references are consistently formatted according to the journal style.
Reviewer 3 Report
Comments and Suggestions for Authors
Dear Author(s):
The reviewer shall know that authors' provided prompts are enough for getting the results which has been claimed the the results from their manuscript.
More importantly, the just demonstration of Generative AI shall not meet the quality of any scientific and/or case journals like Healthcare (again, your samples are too small to say "case").
Comments on the Quality of English Language
Not applicable, the reviewer didn't check the quality of English this time.